A multi-sensor array for detecting and analyzing nocturnal avian migration

Strand Alva I. 1 2 alvastrand@ou.edu
Bridge Eli S. 2
Kelly Jeffrey F. 1 2
Stepanian Phillip M. 3
Bodine David J. 4 5
Soto James R. 2
1 Department of Biology, University of Oklahoma , Norman, Oklahoma , United States
2 Oklahoma Biological Survey, University of Oklahoma , Norman, Oklahoma , United States
3 Department of Civil and Environmental Engineering and Earth Sciences, University of Notre Dame , Notre Dame, Indiana , United States
4 Advanced Radar Research Center, University of Oklahoma , Norman, Oklahoma , United States
5 School of Meteorology, University of Oklahoma , Norman, Oklahoma , United States
Nelson David
Electronic publication date: 2023 Aug 30
Publication date: 2023
Volume: 11
Electronic Location ID: e15622
Received 2022 Aug 15; Accepted 2023 Jun 1
Copyright: © 2023 Strand et al.
Copyright year: 2023
Copyright holder: Strand et al.
License: This is an open access article distributed under the terms of the Creative Commons Attribution License, which permits unrestricted use, distribution, reproduction and adaptation in any medium and for any purpose provided that it is properly attributed. For attribution, the original author(s), title, publication source (PeerJ) and either DOI or URL of the article must be cited.
License URL: https://creativecommons.org/licenses/by/4.0/

Keywords: Avian migration, Data integration, Remote sensing, Weather surveillance radar

Funding: National Science Foundation 1840230 Oklahoma Biological Survey University of Oklahoma Vice President for Research and Partnerships University of Oklahoma Thousands Strong crowdfunding platform This work was supported by the National Science Foundation (award numbers 1840230 to J. F. K., E. S. B., and P. M. S. and 1545261 to J. F. K. and E. S. B.), the Oklahoma Biological Survey, the University of Oklahoma Vice President for Research and Partnerships, and the University of Oklahoma Thousands Strong crowdfunding platform. Financial support was provided by the University of Oklahoma Libraries’ Open Access Fund. The funders had no role in study design, data collection and analysis, decision to publish, or preparation of the manuscript.

==============================
Avian migration has fascinated humans for centuries. Insights into the lives of migrant birds are often elusive; however, recent, standalone technological innovations have revolutionized our understanding of this complex biological phenomenon. A future challenge for following these highly mobile animals is the necessity of bringing multiple technologies together to capture a more complete understanding of their movements. Here, we designed a proof-of-concept multi-sensor array consisting of two weather surveillance radars (WSRs), one local and one regional, an autonomous moon-watching sensor capable of detecting birds flying in front of the moon, and an autonomous recording unit (ARU) capable of recording avian nocturnal flight calls. We deployed this array at a field site in central Oklahoma on select nights in March, April, and May of 2021 and integrated data from this array with wind data corresponding to this site to examine the influence of wind on the movements of spring migrants aloft across these spring nights. We found that regional avian migration intensity is statistically significantly negatively correlated with wind velocity, in line with previous research. Furthermore, we found evidence suggesting that when faced with strong, southerly winds, migrants take advantage of these conditions by adjusting their flight direction by drifting. Importantly, we found that most of the migration intensities detected by the sensors were intercorrelated, except when this correlation could not be ascertained because we lacked the sample size to do so. This study demonstrates the potential for multi-sensor arrays to reveal the detailed ways in which avian migrants move in response to changing atmospheric conditions while in flight.

Introduction

Every spring and fall, billions of birds migrate, and yet, despite this enormous number, large gaps remain in our understanding of avian migration. In fact, avian migration is challenging to study because migratory birds are small and move across vast spatial and temporal scales, often at night. To this end, several remote-sensing technologies have emerged to study the movements of avian migrants (Bridge et al., 2011; McKinnon & Love, 2018; Robinson et al., 2010). One such technology that has been successfully used to unravel the complexities of avian migration is weather surveillance radar (WSR), particularly in the United States, as the spatial and temporal coverage of the U.S. network of WSRs, called NEXt-Generation Weather RADar (NEXRAD, or WSR-88D), is extensive. Additionally, NEXRAD collects data nearly continuously, and these data are openly accessible. Indeed, WSRs have been used to monitor animal activity (Kelly et al., 2017; Kelly & Horton, 2016), extrapolate population estimates (Clark et al., 2020), and identify trends that correlate with land use and climate change (Bridge et al., 2016). However, this technology suffers from three major shortcomings. Firstly, in most cases, WSRs cannot be used to identify species aloft. Secondly, although WSRs have been used to estimate mean flight directions and velocities of migrants (Horton et al., 2016), they cannot detect individual flight characteristics, including individual flight directions, flight velocities, and body orientations. Thirdly, NEXRAD WSRs have a range bias that makes it challenging to detect low-flying birds (Buler & Diehl, 2009; Diehl & Larkin, 2005). Therefore, integrating NEXRAD WSR data with data from other remote sensors that can address these limitations would enhance our ability to study avian migrants.

To demonstrate the utility of doing so, we designed a proof-of-concept multi-sensor array. This array consisted of two WSRs (the KTLX WSR from the NEXRAD network and a portable, dual-polarization X-band WSR called PX-1000 (Cheong et al., 2013)), a novel, automated observational sensor based on traditional moon-watching called LunAero (Honeycutt et al., 2020; Honeycutt & Bridge, 2022), and a microphone capable of recording avian nocturnal flight calls. We deployed this array at the University of Oklahoma Kessler Atmospheric and Ecological Field Station (KAEFS) on select nights in March, April, and May of 2021. Data from this array were then integrated with wind data from the National Centers for Environmental Prediction (NCEP)’s North American Regional Reanalysis (NARR) to investigate the influence of this abiotic variable on avian migrants in flight. We chose to include a microphone, a moon-watching sensor, and a portable WSR in our array because each of these sensors addressed one of the three limitations of NEXRAD WSRs described above. The microphone, coupled with knowledge of avian species’ nocturnal flight calls, allows us to identify species migrating aloft. LunAero, the moon-watching sensor, can be used to derive flight directions of individual birds and provide insights into wind-drift compensation of individual migrants (Honeycutt et al., 2020; Honeycutt & Bridge, 2022). Finally, PX-1000, the portable WSR, can be used to detect local migrants flying at low altitudes (Cheong et al., 2013).

With this multi-sensor array, we sought to answer two major questions: Are the avian migration intensities detected by all of the sensors intercorrelated?

Could integration of sensor data enable inferences about the influence of wind on avian migration?

Firstly, we needed to validate the ability of LunAero and PX-1000 to detect birds, as these are novel sensors. We predicted that the number of individual birds detected by LunAero would be positively correlated with the regional intensity of avian migration from the KTLX radar. We also predicted that the local intensity of avian migration detected by PX-1000 would be positively correlated with the regional avian migration intensity detected by the KTLX radar.

Secondly, we wanted to investigate whether we could use data from the multi-sensor array that we designed and integrate it with wind data from NARR to demonstrate how such a multi-sensor array could be used to study how birds respond to changing wind conditions during spring migration. To do so, we set out to verify that the wind directions and velocities from NARR on the nights in question matched our expectations. We predicted that most winds, including the fastest ones, would be southerly (blowing from south to north), as the Great Plains low-level jet (LLJ) provides strong, southerly winds across our study site in the spring (Wainwright, Stepanian & Horton, 2016). Next, we wanted to use data from the KTLX radar and NARR to identify the wind conditions favored by migrants across select nights in the spring. We predicted that they would favor southerly (tail) winds, as birds migrate from south to north in the spring, in addition to weaker winds.

Importantly, we wanted to leverage data on individual migrants from LunAero and integrate it with data from NARR to determine the ways in which wind conditions influence the flight directions of individual migrants in the spring. We predicted that when exposed to stronger southerly winds, migrants would adjust their flight directions to drift. That is, as the velocity of the wind increases, we predicted that the difference between the direction of the wind and the flight direction of an individual migrant would decrease. We made this prediction for southerly winds, as the direction of those winds is the same as that in which birds migrate in the spring. Finally, we wanted to investigate whether we could integrate avian nocturnal flight call data with data from other sensors to examine whether different avian species favor different wind conditions during migration.

Materials and Methods

Study site

All observations were made from March to May of 2021 in the lower atmosphere above the University of Oklahoma Kessler Atmospheric and Ecological Field Station (KAEFS) (34.98°N, 97.52°W). KAEFS encompasses ~146 ha at ~350 m above mean sea level and features large patches of mixed-grass prairie that provide a full view of the sky.

Temporal sampling

The instruments deployed at KAEFS included an automated moon-watching apparatus, the PX-1000 x-band radar, and an autonomous recording unit (ARU) to record audio data. The moon-watching equipment was deployed only within 5 days of a full moon, and the data were only usable when the sky was clear of clouds. The ARU was deployed throughout the period surrounding the full moon, and the PX-1000 collected data continuously for the entire spring. However, the data used in this paper is restricted to six nights for which we had moon-watching footage. These nights were March 26, March 27, March 28, March 29, April 24, and May 25, 2021. Furthermore, because the ARU did not detect any migrant species on the nights of March 26 and March 28, only ARU data for four nights (the nights of March 27, March 29, April 24, and May 25) could be and were analyzed.

NEXRAD weather surveillance radar (WSR) data

KAEFS is located ~45 km southwest of the nearest NEXRAD weather surveillance radar (WSR), which is the KTLX radar (35.34°N, 97.28°W). To estimate regional intensity of avian migration, data from KTLX were accessed from the Amazon Web Service radar archive (https://registry.opendata.aws/noaa-nexrad/). We opened each radar file using PyART (Helmus & Collis, 2016), censored weather by removing pixels with a depolarization ratio less than −12.5 dB (Kilambi, Fabry & Meunier, 2018), limited the analysis range to within 60 km of the radar, and manually removed clutter from a wind farm west of the radar site. To create vertical profiles of reflectivity (VPRs), we determined the altitude of each remaining radar gate center and assigned the corresponding reflectivity (in cm2/km3) to this altitude. Reflectivities were averaged within 100-m altitudinal bins (starting at 0 m above mean sea level), yielding a single reflectivity value per bin. Reflectivities between 400 and 1,300 m above mean sea level (between ~50 and ~950 m above ground level) were summed and multiplied by 0.1 km to account for altitudinal bin height, yielding a single reflectivity value per VPR in cm2/km2 (Chilson et al., 2012a, 2012b). Reflectivities for each VPR were then summed for each 1-h period, yielding a single summed reflectivity value for each combination of date and hour, hereafter referred to as the regional intensity of avian migration from the KTLX radar for each combination of date and hour.

Wind data

To examine the influence of wind on nocturnal avian migration, we made use of wind data from the National Centers for Environmental Prediction (NCEP)’s North American Regional Reanalysis (NARR) data set. This data set provides weather data based on an interpolation model for 32 km grid cells that encompass North America. NARR data have a temporal resolution of 3 h. We downloaded wind data from the NCEP’s Global Reanalysis page (https://psl.noaa.gov/data/gridded/data.narr.html) and extracted wind velocity (in meters per second) and direction (in degrees clockwise from north) from the grid cell overlapping KAEFS (35.18°N, 97.44°W). Each wind profile was integrated across altitudes by calculating the mean wind velocity and direction between 400 and 1,300 m above sea level.

Moon-watching (LunAero) data

Moon-watching is a method of quantifying nocturnal bird migration, wherein one uses the moon as an illumination source to count the silhouettes of birds that fly between the moon and the observer. Although moonwatching has been employed by researchers on several continents (Hilgerloh, Weinbecker & Zehtindjiev, 2006; Liechti, Bruderer & Paproth, 1995; Lowery, 1949; Lowery & Newman, 1966; Nisbet, 1959; Trösch et al., 2005; Weisshaupt, Maruri & Arizaga, 2016; Zehtindjiev & Liechti, 2003), more widespread use has been limited by challenges relating to accuracy of the method and the observational rigor required when collecting data manually in real time. We used the LunAero automated moon-watching sensor to observe bird migration by making video recordings of the moon and identifying bird flight paths using computer vision software (Honeycutt et al., 2020; Honeycutt & Bridge, 2022).

The analysis workflow for extracting migratory bird information from videos is summarized below. All analyses were done in the R programming environment (R Core Team, 2022) and used Stellarium, an open-source planetarium (Zotti et al., 2021). Briefly, the analysis pathway was as follows: We used computer vision tools available in Rvision (Garnier & Muschelli, 2022), which is a wrapper for OpenCV (Bradski, 2000) to identify dark pixel clusters (henceforth contours) in each frame of video using the dynamic thresholding function. The contours identified by this process included both bird silhouettes and a considerable amount of noise associated with camera movement and dark features on the moon. We applied some filtering based on co-occurrence across frames to remove contours that were likely to be due to moon craters, and we ranked contours according to how distinct they were from the immediate background (more distinct contours were more likely to be actual bird silhouettes).

The next analysis step examined all of the extracted contours and attempted to assemble them into linear series of points across a series of video frames. Each contour not already assigned to a track was compared with all contours from three previous and three subsequent frames to find potential flight paths. If a series of at least three contours was detected that met a threshold for linearity, the search for contours was extended to add to the potential flight path. Any contours added to a flight path were flagged such that they could not contribute to another flight path. The results of this step generally yielded a majority of real flight paths, but instances where paths were assembled from false contours were frequent. These false detections were usually evident when filtering examining by the linearity of the contours, the number of contours comprising a path, and the distinctiveness of the contours.

To remove false paths, we visually inspected sets of video frames associated with paths that were questionable as indicated by low contour counts, reduced linearity, and very faint contours. We also visually inspected distinct contours that were not linked to be a flight path to ensure that we did not fail to detect any birds.

When collecting video footage, we did not ensure that the recorded images were aligned with true horizontal, which would affect assignment of flight directions. To correct for potentially rotated video footage, we aligned moon images from our video with vertically aligned images from the Stellarium software package (Zotti et al., 2021). The rotation factor required for this alignment was used to adjust the x and y coordinates of the flight paths to correct for the misalignment of the camera with the horizontal plane.

We transformed the x and y coordinates to situate them on a horizontal plane parallel to the earth’s surface and we rotated them such that the direction of each flight path could be measured in degrees east of north.

PX-1000 weather surveillance radar (WSR) data

The PX-1000 radar was collocated with the moonwatching equipment at KAEFS. The PX-1000 is a transportable, solid-state, polarimetric radar with independent horizontal and vertical polarizations. The radar scanned at a frequency of 9,550 MHz at elevation angles of 4 to 10 degrees at two-degree increments, while rotating 360 degrees every 20 s. The PX-1000 recorded data continuously throughout each night of data collection. However, we used different radar pulse settings in May compared to March and April. In March and April, we used a pulse duration of 67-microseconds, which is typical for use in scanning long range targets (e.g., objects more than 5 km away). Using pulse compression, range resolution of 30 m is obtained with these settings (Kurdzo et al., 2014). The longer pulse lengths involve emission of more radar energy on average, which makes the radar sensitive to more distant scatterers, but it limits the extent to which we can resolve reflectivity from objects that are close to the radar (e.g., within 10 km). In May, a new pulse compression technique was used that involves simultaneous transmission and reception and increases sensitivity at close range by ~10 dB (Aquino, Cheong & Palmer, 2021). Mirroring our method of processing the KTLX radar data, we obtained summed reflectivity values for each 10-min time period, hereafter referred to as local intensity of avian migration from the PX-1000 radar for each combination of date and time period.

Nocturnal flight call recorder and detection

We recorded nocturnal flight calls with a pressure zone microphone (Wildtronics Micro Mic PIP Microphone, Newton Falls, Ohio, USA) attached to an Audiomoth Autonomous Recording Unit (ARU) (Hill et al., 2019). The microphone and support apparatus were constructed following the OldBird 21 c Pressure Zone Microphone design developed by Bill Evans (http://www.oldbird.org/mic/21c.htm). The microphone was positioned at the apex of an inverted plexiglass pyramid designed to increase the gain of audio recordings. The pyramid was supported by a plexiglass pedestal that rested inside a plastic two-gallon bucket with the microphone pointing upward to the sky. The bucket served to minimize background sounds (e.g., insects) from below. Plastic wrap was taped in place over the pyramid and pedestal to provide a waterproof cover for the microphone. The Audiomoth was placed in a waterproof container attached to the bottom of the bucket with the microphone cord inserted through it.

We recorded avian nocturnal flight calls from March 25, 2021 to May 31, 2021; however, we restricted our analysis of the ARU data to six nights in March, April, and May for which there was LunAero data (see Temporal Sampling). Furthermore, because the ARU did not detect any migrant species on the nights of March 26 and March 28, only ARU data for four nights (March 27, March 29, April 24, and May 25) could be and were analyzed. Each day, recording started at 2000 Central Time (0100 UTC) and ended at 0700 Central Time (1200 UTC). We analyzed audio recordings with the Vesper Audio Analysis Program (Harold Mills Revision aa3ad038) which processes the audio recordings then generates spectrograms of brief audio segments that the program identifies as potential flight calls. Vesper generates these audio segments or “clips” using multiple detectors that scan the entire audio for specified frequencies and amplitudes.

We examined all of the clips generated by Vesper analysis and assigned them to species by visually comparing clip spectrograms to reference spectrograms generated from Flight Calls of Migratory Birds (Evans & O’Brien, 2002), Rosetta Stone to the Warblers (Farnsworth, 2011), studies of various call complexes (Landsborough, Foote & Mennill, 2019), and personal recordings of known species observed in the field. We used Raven Pro Software Version 1.6.3 (Cornell University, 2022) to measure length of calls, determine specific frequencies of calls, measure distance between modulation peaks of certain calls, and listen to calls and the entirety of the audio recordings when needed. We compiled data for all species detected and all flight calls. We then filtered out all non-migratory species and resolved all calls that were likely repeated vocalizations from the same individual into a single detection based on the timing and relative volume of the recorded vocalization.

Statistical analysis

To validate the ability of the LunAero moon-watching sensor to detect birds, we compared the number of individual birds detected by LunAero with the regional intensity of avian migration from the KTLX radar for each of six nights in March, April, and May of 2021 and each hour for which there was data. That is, we calculated both the number of individual birds detected and the regional migration intensity for each combination of date and hour, matched these numbers with one another based on the date and hour, and calculated Pearson’s correlation coefficient as a measure of the strength and direction of the correlation between these two variables across these nights.

To validate the ability of the PX-1000 radar to detect birds, we compared the local intensity of avian migration from PX-1000 with the regional intensity of avian migration from the KTLX radar by matching each migration intensity value of KTLX (for a ~10-min time period and one of the six nights in March, April, and May of 2021) to the migration intensity value of PX-1000 closest in time to it (also calculated for a 10-min time period). Then, we calculated Pearson’s correlation coefficient as a measure of the strength and direction of the correlation between these two variables across nights.

To showcase how this multi-sensor array could be used to study the influence of wind on avian migration, we compared regional migration intensity data from KTLX and wind data from NARR by matching each migration intensity value of KTLX (for a ~10-min time period and one of the six nights in March, April, and May of 2021) to the wind velocity value closest in time to the migration intensity value. Then, we calculated Pearson’s correlation coefficient as a measure of the strength and direction of the correlation between these two variables across nights.

Finally, to demonstrate the utility of LunAero as a tool to study the individual movements of birds during migration, we compared flight directions of individual migrants detected by LunAero with those of winds from NARR by, once again, matching each individual bird detection by LunAero with the wind velocity and direction values closest in time to it. Finally, we calculated Pearson’s correlation coefficient as a measure of the strength and direction of the correlation between these two variables across nights.

Results

To validate the ability of the LunAero moon-watching sensor to detect birds, we compared the number of individual birds detected by LunAero with the regional intensity of avian migration from the KTLX radar. We found that these two variables are positively and statistically significantly correlated with each other on five nights in March and April of 2021 (r(31) = 0.714, p < 0.001), although the strength of this correlation decreases and the correlation ceases to be statistically significant with the inclusion of data collected on the night of May 25, 2021 (between 10 PM and 1 AM) (r(34) = 0.322, p = 0.0553) (Fig. 1).

Figure 1 Relationship between the number of birds detected by LunAero and the regional avian migration intensity from KTLX in the atmosphere above KAEFS on select nights in March, April, and May of 2021.

Each data point corresponds to a 1-h period on the night of March 26, March 27, March 28, March 29, April 24, or May 25, 2021. Each date is associated with a different color.

To validate the ability of PX-1000 to detect birds, we compared the local intensity of avian migration from PX-1000 with the regional intensity of avian migration from the KTLX radar. We found that the strength of the positive correlation between these two measures of migration intensity hinges on the way in which the PX-1000 radar data is processed. In May, the way in which the PX-1000 data was processed was modified to increase sensitivity at close range. Consequently, we found that the correlation between the local migration intensity from PX-1000 and the regional migration intensity from the KTLX radar is strong and statistically significant for the night of May 25, 2021 (r(168) = 0.768, p < 0.001; Fig. 2A) but weak and not statistically significant across five nights in March and April (r(320) = −0.0977, p = 0.0799; Fig. 2B).

Figure 2 Relationship between the local avian migration intensity detected by PX-1000 and the regional avian migration intensity detected by KTLX in the atmosphere above KAEFS in the spring of 2021.

(A) Relationship on the night of May 25, 2021. (B) Relationship on the nights of March 26, March 27, March 28, March 29, and April 24, 2021. Each data point corresponds to a ~10-min period, which is the temporal resolution of the KTLX weather surveillance radar (WSR). The regional migration intensity on the night of May 25, 2021 significantly predicts and explains 59.1% of the variance in local migration intensity (β = 5.00 ± 0.321, P < 0.001).

Next, we analyzed wind data from NARR to confirm that the wind conditions on the nights of March 26, March 27, March 28, March 29, April 24, and May 25, 2021 matched our expectations. We found that most of the winds were southerly (blowing from south to north), as expected (Fig. 3). Furthermore, we found that the fastest winds were southerly, as expected (Fig. 3).

Figure 3 Direction and velocity of the wind in the atmosphere above the Kessler Atmospheric and Ecological Field Station (KAEFS) on select nights in March, April, and May of 2021.

This polar plot shows wind direction as the angle in degrees clockwise from north and velocity as the distance between the data point and the origin. Each data point corresponds to a 3-h period for a given date, represented by a color. This data stems from the National Centers for Environmental Prediction (NCEP)’s North American Regional Reanalysis (NARR).

We also used regional avian migration intensity data from the KTLX radar and wind data from NARR corresponding to these dates in the spring of 2021 to determine the wind conditions favored by migrant birds across these dates, demonstrating the use of this multi-sensor array as a tool to study avian migration. We found a statistically significant, negative correlation between regional migration intensity and wind velocity, suggesting that migrant birds favor weaker winds (r(490) = −0.259, p < 0.001; Fig. 4).

Figure 4 Relationship between regional avian migration intensity detected by KTLX and wind velocity in the atmosphere above KAEFS on select nights in March, April, and May of 2021.

Each data point corresponds to a ~10-min period, which is the temporal resolution of the KTLX weather surveillance radar (WSR), for the nights of March 26, March 27, March 28, March 29, April 24, and May 25 of 2021. The wind data stems from the NCEP’s NARR.

We also compared the flight directions of individual migrant birds detected by LunAero with those of winds from NARR to understand the ways in which migrant birds respond to changing wind conditions across these dates. For southerly winds, we found that as the velocity of the wind increases, the difference between the direction of the wind and the flight direction of the individual migrant decreases and that this negative correlation is statistically significant (r(1153) = −0.284, p < 0.001; Fig. 5A). We did not find a statistically significant correlation between wind velocity and the difference between wind and flight directions for northerly winds (r(868) = 0.0686, p = 0.0415; Fig. 5B).

Figure 5 Difference between the flight direction of individual birds detected by LunAero and the direction of the wind as a function of wind velocity in the atmosphere above KAEFS in the spring of 2021.

(A) Southerly and (B) northerly winds. Southerly winds are winds with direction angles between 90 and 270 degrees clockwise from north, while northerly winds are winds with direction angles between 270 and 0 degrees and between 0 and 90 degrees clockwise from north. The black dots represent mean differences for discrete wind velocity values. The data correspond to the nights of March 26, March 27, March 28, March 29, April 24, and May 25, 2021. The wind data stems from the NCEP’s NARR.

Finally, we used nocturnal flight call recordings to analyze the avian migrant species composition of the lower atmosphere above the University of Oklahoma Kessler Atmospheric and Ecological Field Station (KAEFS) on four nights in March, April, and May of 2021. We recorded the largest number of individual migrants on the night of April 24 (41 individual birds and 11 species) and the second-largest number of individual migrants on the night of May 25 (19 individual birds and three species) (Table 1). Notably, we recorded 14 Upland Sandpipers on the night of April 24 and 17 Swainson’s Thrushes on the night of May 25 (Table 1). Due to the small sample size of the nocturnal flight call data, we could not make inferences about the relationship between the number of migrants recorded and the number of migrants detected by LunAero and the relationship between nocturnal flight calls and wind conditions during migration.

Table 1 Avian species and number of individuals detected via nocturnal avian flight calls in the atmosphere above KAEFS on select nights in March, April, and May of 2021.

	Scientific name	Common name	March 27, 2021	March 29, 2021	April 24, 2021	May 25, 2021	Number of individuals per species	
	Protonotaria citrea	Prothonotary Warbler	1	0	0	0	1	
	Passerculus sandwichensis	Savannah Sparrow	3	0	4	0	7	
	Branta canadensis	Canada Goose	0	1	0	0	1	
	Calidris melanotos	Pectoral Sandpiper	0	1	0	0	1	
	Spizella pallida	Clay-colored Sparrow	0	0	4	0	4	
	Spizella passerina	Chipping Sparrow	0	0	1	1	2	
	Ammodramus savannarum	Grasshopper Sparrow	0	0	5	0	5	
	Tringa flavipes	Lesser Yellowlegs	0	0	1	0	1	
	Catharus ustulatus	Swainson’s Thrush	0	0	4	17	21	
	Bartramia longicauda	Upland Sandpiper	0	0	14	0	14	
	Pooecetes gramineus	Vesper Sparrow	0	0	2	0	2	
	Zonotrichia leucophrys	White-crowned Sparrow	0	0	3	0	3	
	Cardellina pusilla	Wilson’s Warbler	0	0	2	0	2	
	Zonotrichia albicollis	White-throated Sparrow	0	0	1	0	1	
	Setophaga petechia	Yellow Warbler	0	0	0	1	1	
Number of individuals per day			4	2	41	19	66	
Note:

Nocturnal avian flight calls were recorded by an autonomous recording unit (ARU) on the nights of March 27, March 29, April 24, and May 25, 2021.

Discussion

The challenges of studying avian migration have spurred the development of a number of remote-sensing technologies to detect migrant birds. As each of these technologies have their strengths and weaknesses, efforts have been made to use them in tandem; however, a majority of these efforts have been limited to two technologies ((Gauthreaux & Belser, 1998; Larkin, Evans & Diehl, 2002; Weisshaupt, Lehtiniemi & Koistinen, 2021) or to radar technologies only (Liechti et al., 2019; Nilsson et al., 2018), although researchers have increasingly been exploring the utility of combining more than two different types of technologies (Liechti, Bruderer & Paproth, 1995; Weisshaupt et al., 2017)). We designed a proof-of-concept multi-sensor array, integrating data from multiple technologies to detect migrants in flight and elucidate the ways in which birds respond to changing wind conditions during spring migration.

We found a strong, statistically significant, positive correlation between the number of birds detected by LunAero and the regional intensity of avian migration detected by the KTLX radar on five nights in March and April of 2021, validating the ability of LunAero to detect birds (Fig. 1). However, this correlation decreased and ceased to be statistically significant with the inclusion of data collected on the night of May 25, 2021 (Fig. 1). It is unclear why the relationship between these two variables differed on this night. While it is possible that LunAero is unable to detect birds flying above a certain altitude, we performed an analysis that suggests that migrant altitude does not explain why LunAero did not detect many birds on that night. In fact, we calculated the mean altitude at which birds were detected by the KTLX radar on each of the six night in March, April, and May, and we found that the mean altitude at which birds were detected on May 25, 2021 was not the highest (Table 2). It is also possible that the body size of the migrants on the night of May 25 was smaller, thus making it more challenging for LunAero to detect them. In fact, evidence suggests that the mean body mass of the assemblage of avian migrants decreases from April to May (Horton et al., 2018). Still, it appears that on most nights, LunAero can detect many of these migrants and can be used to study their movements. In fact, LunAero may be able to detect low-flying migrants that may be challenging to detect using weather surveillance radars (WSRs). It is also important to note that we used raw counts to estimate the number of birds detected by LunAero and did not account for the fact that the volume of the atmosphere sampled by LunAero is dependent on the elevational angle of the moon (i.e., how close the moon is to the horizon). In fact, LunAero samples a larger volume of the atmosphere, thus detecting more birds, at lower moon elevational angles (i.e., when the moon is closer to the horizon) (Lowery, 1949). In the future, we will improve the ability of LunAero to detect birds by accounting for the contribution of the moon elevational angle.

Table 2 Mean altitude at which birds were detected by the KTLX radar on six nights in March, April, and May of 2021.

Date	Mean altitude above mean sea level (m)	
March 26, 2021	770.210404	
March 27, 2021	537.570479	
March 28, 2021	667.877065	
March 29, 2021	749.929377	
April 24, 2021	689.156399	
March 25, 2021	733.208123	

We also found a strong, statistically significant, positive correlation between the local migration intensity detected by the PX-1000 radar and the regional migration intensity detected by the KTLX radar on the night of May 25, 2021, preliminarily validating the ability of the PX-1000 radar to detect birds (Fig. 2A). The correlation between these two measures of migration intensity was weak and not statistically significant across five nights in March and April, but that is because the sensitivity of the PX-1000 radar was lower at close range in March and April than in May (Fig. 2B). As such, given its current processing pipeline, the PX-1000 radar could be deployed to fill the gaps left by the NEXRAD network: where there is no or low spatial NEXRAD coverage or to study the movements of low-flying migrants that may be challenging to detect using NEXRAD WSRs. Therefore, the PX-1000 could be used to paint a more comprehensive picture of local-scale movements of migrants in conjunction with NEXRAD WSRs.

By integrating regional migration intensity data from the KTLX radar with wind data from NARR, we found evidence suggesting that birds prefer to migrate under weak wind conditions in the spring (Fig. 4). This finding is consistent with research that suggests that conditions with low turbulence promote flight efficiency, particularly for smaller avian species (Bowlin & Wikelski, 2008). Additionally, by integrating flight directional data for individual migrants from LunAero with wind data from NARR, we found that when migrating under strong, southerly wind conditions across six nights in March, April, and May of 2021, birds take advantage of these conditions by adjusting their flight direction to match that of the wind (Fig. 5A). This finding is consistent with previous research showing that birds take advantage of favorable wind conditions during migration (Horton et al., 2016, 2018). Our sample size is low (there were only six nights for which there was LunAero data), so we cannot extrapolate these results across the entirety of the spring season (or across spring seasons), but given the fact that our findings match those in the literature, these analyses function as both another way to partially validate the sensors in our array and to showcase the potential of multi-sensor arrays to reveal how birds respond to changing wind conditions during migration.

We were hoping to be able to integrate avian nocturnal flight call data with data from other sensors in our array; unfortunately, we were unable to collect enough nocturnal flight call data to do so. We recorded the largest numbers of migrants on the nights of April 24 and May 25, which were also the nights with the highest regional migration intensities detected by the KTLX radar, and yet, despite this fact, we only recorded 41 and 19 individuals on the nights of April 24 and May 25, respectively. It is possible that the detection range of the microphone used to record nocturnal flight calls is limited, as evidence suggests that detection rates of warbler nocturnal flight calls are below 50% within the first 50 m of elevation and below 25% within the first 100 m with a temperature of 20 degrees Celsius, a relative humidity of 50%, and a pressure of 1,013.25 hPa (Horton et al., 2015). Although the detection range of flight calls is limited, they provide unique insight into species identities that are not possible to infer from other data sources.

Our study demonstrates that multi-sensor arrays can be successfully deployed not only to detect migrant birds in flight, but to understand the ways in which these birds move in response to changing environmental conditions. It also sheds light on the challenges that come with successfully deploying such arrays. The array we designed consisted of four different sensors whose data were integrated with data from NARR, yielding five different data sources. As the number of sensors and, thus, the number of data sources increases, integrating these data becomes more challenging. In fact, the spatial and temporal scales of these data did not always match. For example, integrating the avian nocturnal flight call data with the rest of the data was challenging because we only recorded migrant flight calls on four of six nights, and when we did record such calls, we recorded them infrequently over the course of those nights. Mismatches in spatial and temporal scales has been recognized as a major methodological challenge of data integration in macrosystems ecological research, as mismatched data need to be up- or down-scaled for the extent and/or resolution to match (Rüegg et al., 2014; Zipkin et al., 2021). Therefore, we recommend creating pre- and post-integration plans prior to deploying multi-sensor arrays to make sure that it is possible to integrate the data in the first place and that there is a contingency plan if the spatial and/or temporal scales of the data change or one or more of the sensors fails.

Another major challenge of successfully deploying multi-sensor arrays is interdisciplinary collaboration, as the collection, integration, and analysis of data from different sources requires the collaboration of people from different disciplines (Rüegg et al., 2014). In our case, we required knowledge in areas of study as far-ranging as radar engineering, computer vision, and flight call identification. However, despite the challenges that come with successfully deploying multi-sensor arrays, doing so within an interdisciplinary framework would allow us to advance our understanding of avian migration at the macrosystem scale (Kelly & Horton, 2016) and should be encouraged. The sample size of the data retrieved from our array was small, but that is because it was meant to be a proof-of-concept array deployed within narrow spatial and temporal extents; increasing the extent and resolution of the data would enhance the power of such an array to detect movements of migrants over the course of a season within a region across altitudes. As such arrays expand, it will become increasingly important for interdisciplinary teams to adopt rigorous data management practices (Rüegg et al., 2014). As we look to the future, we hope to see multi-sensor arrays being used to provide a more comprehensive picture of avian migration at local, regional, and continental scales.

Conclusions

We designed a proof-of-concept multi-sensor array to study nocturnal avian spring migration in central Oklahoma. This array, which consisted of a NEXRAD weather surveillance radar (WSR), a portable X-band WSR, a moon-watching sensor (LunAero), and an autonomous recording unit (ARU), was deployed at the University of Oklahoma Kessler Atmospheric and Ecological Field Station (KAEFS) on six nights in March, April, and May of 2021. We verified that the sensors agreed with one another, that is, that the intensities of avian migration recorded by most of these sensors were intercorrelated. We were unable to verify that the number of individual birds recorded by the ARU was consistent with the avian migration intensities detected by the other sensors due to the small sample size of nocturnal avian flight calls. We also integrated data from this array of sensors with wind data from the National Centers for Environmental Prediction (NCEP)’s North American Regional Reanalysis (NARR) to examine the influence of this abiotic variable on the movements of avian migrants aloft across these six nights in the spring. We found that the number of birds detected by LunAero increased as wind velocity decreased, indicating that birds prefer to migrate when winds are weak. In addition, we found that when exposed to strong, southerly winds across these six nights, avian migrants took advantage of these conditions by aligning their flight direction with that of the wind (i.e., drifting). Continued development of the moon-watching sensor would allow us to study additional features of individual migrants, such as flight velocity and body orientation, shedding light on the ways in which avian migrants respond to atmospheric conditions in flight. Ultimately, expanding the temporal and spatial scales at which this multi-sensor array operates holds promise for the study of avian migration by providing a fuller picture of this phenomenon in time and space.

Supplemental Information

Supplemental Information 1 Jupyter notebook with all of the code used to analyze raw data.

Click here for additional data file.

Supplemental Information 2 Velocity profiles of reflectivity (VPRs) from the PX-1000 radar.

A VPR was created for each volume scan from 0 to 12 UTC on 3/26/2021, 3/27/2021, 3/28/2021, 3/29/2021, 4/24/2021, and 5/25/2021.

Click here for additional data file.

Supplemental Information 3 North American Regional Reanalysis (NARR) wind profiles.

A wind profile was created for each three-hour interval between 0 and 12 UTC on 3/26/2021, 3/27/2021, 3/28/2021, 3/29/2021, 4/24/2021, and 5/25/2021.

Click here for additional data file.

Supplemental Information 4 Moon-watching data collected at Kessler Atmospheric and Ecological Field Station on the nights of 3/26/2021, 3/27/2021, 3/28/2021, 3/29/2021, 4/24/2021, and 5/25/2021.

Each row denotes the flight path of a bird.

Click here for additional data file.

Supplemental Information 5 Avian nocturnal flight calls recorded on the nights of 3/26/2021, 3/27/2021, 3/28/2021, 3/29/2021, 4/24/2021, and 5/25/2021 at Kessler Atmospheric and Ecological Field Station.

Each row denotes a flight call.

Click here for additional data file.

We thank the University of Oklahoma Advanced Radar Research Center engineers and technicians for maintaining the PX-1000 radar and providing support during data collection. In particular, Boon Leng Cheong assisted the authors with implementing scanning strategies. We also thank Wesley T. Honeycutt, Angela Chen, Riley Miller, Israel Lugo, and Alyse V. Heaston for their roles in the development of the LunAero hardware and analysis pipeline.

Additional Information and Declarations

Competing Interests

Author Contributions

Data Availability

The authors declare that they have no competing interests.

Alva I. Strand conceived and designed the experiments, analyzed the data, prepared figures and/or tables, authored or reviewed drafts of the article, and approved the final draft.

Eli S. Bridge conceived and designed the experiments, performed the experiments, analyzed the data, authored or reviewed drafts of the article, and approved the final draft.

Jeffrey F. Kelly conceived and designed the experiments, analyzed the data, authored or reviewed drafts of the article, and approved the final draft.

Phillip M. Stepanian conceived and designed the experiments, analyzed the data, authored or reviewed drafts of the article, and approved the final draft.

David J. Bodine conceived and designed the experiments, performed the experiments, analyzed the data, authored or reviewed drafts of the article, and approved the final draft.

James R. Soto conceived and designed the experiments, performed the experiments, analyzed the data, authored or reviewed drafts of the article, and approved the final draft.

The following information was supplied regarding data availability:

The raw data, including the Jupyter notebook with all of the code used, the data from the PX-1000 radar, the data from NARR, the moon-watching data, and the acoustic data, are available in the Supplemental Files.

NEXRAD KTLX radar data are available at Figshare:

Strand, Alva (2023). VPRs (KTLX radar). figshare. Dataset. https://doi.org/10.6084/m9.figshare.20485737.v1.

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
