# Peer review of "A multi-sensor array for detecting and analyzing nocturnal avian migration"

_PeerJ, doi:10.7717/peerj.15622_

## Round 0.1 · original submission · Major Revisions

Please have a careful look at the comments of both reviewers. They both agree that the paper is interesting and could be suitable for publication in PeerJ, but they also bring up a number of issue that should be addressed. In particular, please consider the comments calling for re-analysis of some of the data, better representation and scope of cited literature, and clearer research goals.

Reviewer 1 ·

Basic reporting

Structure and Criteria
• Professional English language is used throughout.
• The introduction exhibits several shortcomings in the background literature and irrelevant information.
• Literature should be reviewed and relevant papers added which actually fit the content of the text.
• Structure conforms to PeerJ standards.
• Figures’ relevance needs to be reconsidered and labels and quality needs to be improved in almost all figures.
• Raw data is supplied.

Experimental design

• Original primary research within Scope of the journal.
• Research questions are stated clearly, though they need to be revised.
• Investigation suffers from several shortcomings in design.
• Methods should be described more thoroughly.

Validity of the findings

• The work requires a thorough rewriting and preferably also reanalyses and complementary analyses to optimize the outcomes of this work.
• All underlying data have been provided.
• Conclusions need to be revised.

Additional comments

The manuscript needs a thorough revision and partly reanalysis or addition of analyses. There are, however, valuable findings included, so it is worth implementing the required amendments.

Annotated reviews are not available for download in order to protect the identity of reviewers who chose to remain anonymous.

Reviewer 2 ·

Basic reporting

This study compared different remote sensing methods to assess migratory behaviour and avian migrant abundance. I think this paper was very well written and clear. It is self contained with relevant results and objectives. The references are also sufficient, but see a couple of minor comments in the introduction below.

Line 26: Please define NEXRAD it is not term everyone knows.

Line 41: I think this sentence could be enhanced and help it differentiate from the following sentence by adding the words “at the level of the individual at the end of the sentence”.

There are also more recent studies that could be cited here, 2011 is a long time ago. E.g. McKinnon and Love, 2018, Auk. Ten Years tracking the migrations of small landbirds: lessons learned in the golden age of bio-logging

Line 44: This is a great sentence, but I think it needs to be clarified that remote sensing does not replace biologging, it tells researchers different information. Essentially the “therefore” does not entirely make sense here as the jump between individual level tracking and using remote sensing are like comparing apples and oranges. I think perhaps a sentence earlier in this paragraph may help make this distinction clearer that the study of avian migration is a broad field that occurs at multiple scales, and while individual level tracking with biologgers tells us some things and has limitations, another method that tells us different things involves remote sensing. As an alternative, I think the paragraph could also be started with Line 43/44 and may avoid confusing a reader.

Line 50: What does NEXRAD stand for? Please write it out since this is the first instance.

Line 305: Is there a reason “Effort” is capitalized?

Experimental design

Everything was well defined and clear. The methods were described well and would be easy to replicate.

Validity of the findings

Everything has been well reported and the figures and conclusions are well stated/shown and related to the objectives and methodology.

---

## Round 0.2 · Major Revisions

I apologize for the delay in making a decision on your manuscript; the prior handling editor was unable to continue to handle your manuscript and I have just now gotten myself up to speed on the manuscript and the prior reviews.

The manuscript has been re-reviewed by one of the original reviewers. As you can see, that reviewer has concerns/questions about some of the analyses, as well as how some of the results are interpreted. Please address each suggestion of the reviewer in your response.

Reviewer 1 ·

Basic reporting

• Literature is mostly adequate but needs some revision.
• Figures are partly relevant but should be revised according to comments on content.

Experimental design

• Research questions were defined clearly but need reconsideration.
• Low sample size.
• Methods require more detail & information to replicate.

Validity of the findings

- Statistics applied need to be described more in detail

Additional comments

More specific comments on each section:
Abstract
Text has improved considerably.
Introduction
Text has improved considerably, but please consider comments on the wind part (see below). The wind part in its present form is a different kind of research question which does not fit into the framework of device validation.
Methods
A chapter on the statistical analyses is missing which you performed for the comparisons of the devices and the winds. You just stated the Pearson correlation coefficients and p values in the Results.
L113: You state that all observations were from 2021, but in the data table for moonwatching there are observations until 2018. Is this correct? What did you use this data for here? I cannot find this older data in the manuscript. Please include only observations relevant for this study.
L120-127: This Temporal Sampling should be revised. I find it very hard to follow how many nights you actually included and which periods were used for specific combinations of devices. You write several times you used data from March to May, which implies two months of data, which is misleading, because in the end you just used six days (or four?) in this period. You state “most of the data” is restricted to six specific days, even though some sampling happened throughout spring (PX) and later you say acoustic data is just from four days. What periods do you really use for your correlations? Are there different time periods for some comparisons/analyses? Parts of the Results (see comment below) should be shifted to the Methods and complemented by additional information I point out to later. Six days is a very low sample size to draw conclusions on device performance.
L131: What is the lowest height the WSR measures at 45 km, i.e. where the other devices are located?
L149-159: I have a general problem with this wind analysis. Your study is a proof of concept, okay, but you state you want to examine the influence of wind on nocturnal avian migration. However, the informative power of this wind analysis with six randomly distributed nights is negligible. If you want to examine wind effect, why not using the whole period for which you have WSR data. There is quite some literature available on the topic involving radar and wind. Otherwise, for this short period, I would find it more logical, if you compared the directions obtained by the different methods, i.e. do they give the same results? The comparison of the devices is the objective of this paper, as I understand. I would find such a comparative analysis more obvious than one device vs. wind. And I do not understand this north-south classification of winds. If you have winds of e.g. 89 and 91 degrees, you say that one is north and the other south, even though they point east in almost the same direction. To me, a more understandable classification of winds would be e.g. north=[315,45] and south=[135,225]. I suggest removing this wind part and focus on the comparisons of the devices and observations.
L219: I would expect you want to keep the settings stable during a calibration. What is the reason for changing?
L244: You say you obtained calls from March 25 to May 31, which sounds like two months of data, but later you state you used calls from four nights and not even from entire nights? This is misleading, please clarify how many nights you really recorded and how long per night. What is your temporal resolution for the comparisons and why did you just record on four (partial?) nights and not more?
Results
There are several parts in Results, where you somehow introduce the comparisons you performed, e.g. L266-269, 277, 286, 299, but this should be already mentioned in the Methods, please see previous comment on Methods, and then just list more or less directly the results of your comparisons. Please be more specific on the dates you actually used for comparison, “for each combination of date and hour for which there was data” is not very informative. Are these now the six days?
L266+ You just compare intensities from PX-1000 vs KTLX and KTLX vs. LunAero. What about the comparison of PX-1000 and LunAero?
L274-276: Both Pearson correlation coefficients are mentioned twice. Please rephrase this sentence to state it once with the p value.
L277+: Is there any bias from the height range that both radars capture?
How did you get these hourly correlations - did you take all the hourly data points and made one correlation with all of them? I cannot find any information on that. I find it problematic to throw all the hourly data points in one pot and calculate one correlation coefficient. There will be autocorrelation in the hourly data as one hour influences the next one, while nights are independent samples. Furthermore, there can be delays in the flux between sites. I suggest calculating correlations for hourly data for each night separately (i.e. six? correlation coeff), as you did for May 25. Correlation for nightly mean intensities (one correlation coeff) unfortunately does not make sense given the low sample size.
L300+: what kind of analysis did you do for the winds? Some kind of regression? Please specify in Methods.
L307-313: The nights when you recorded should be stated in Methods and here you should show the results.
Discussion
L322-326: If I understand correctly, you refer to remote-sensing technologies and not observational methods in general? Or do you mean observational methods in general? Gauthreaux & Belser 1998 use radar and two observational methods (day observations by binoculars/telescope and at night moonwatching), so one remote sensing and two non-remote sensing approaches. Weisshaupt et al. 2021 compares WR and citizen science observations, i.e. one remote sensing technology and one non-rs. Larking et al. the same. Please clarify what you mean. Then, I am actually not sure, if “most used two technologies” is appropriate. As you state correctly, Liechti et al. and Nilsson et al. use more than two, but also Weisshaupt et al. 2017 (Methods in E&E) and Liechti et al 2019 (Ecography) is a collection of various tools etc. I’d rather say it is “some use two… some…”.
L330-335: How many nights are included for this “strong significant” result? Apparently less than six? The addition of just one night made it non-significant? That does not sound very reassuring.
If I understood well, the KTLX provides height information, so what about height distributions of birds? You just mention intensities, but could flight altitudes explain why the calls performed so badly?
L342-345: I am just wondering, isn’t there common knowledge on migrant composition and thus sizes in different periods of migration seasons?
L355-58: Do I understand correctly, you take one night and find it is significant? I do understand that your manuscript is a proof of concept but still you cannot validate devices by comparing one night.
L366-378: Please mention here your low sample size of six? days.
L416: It would be good to address sample size and discuss its possible effect on outcomes already before. I would expect it to affect all of your comparisons.
References
L495: the DOI is somewhat strange, please check
L569: Incomplete reference: Trösch, B., Lardelli, R., Liechti, F., Peter, D., & Bruderer, B. (2005). Spatial and seasonal variation in nocturnal autumn and spring migration patterns in the western Mediterranean area: A moon-watching survey. Avocetta, 29:63-73. See https://www.avocetta.org/articles/vol-29-2-vju-spatial-and-seasonale-variation-in-nocturnal-autumn-and-spring-migration-patterns-in-the-western-mediterranean-area-a-moon-watching-survey/

Figures
Figures 1 and 2: what about the comparison of PX-1000 and LunAero?
Remove (or replace) the figures on the wind analyses in accordance with previous comments on that part.

---

## Round 0.3 · accepted · Accept

Thanks for addressing the reviewer comments. I believe the manuscript is suited for publication.